# Model-Based Residual Stress Design in Multiphase Seamless Steel Tubes

**DOI:** 10.3390/ma13020439

**Published:** 2020-01-16

**Authors:** Silvia Leitner, Gerald Winter, Jürgen Klarner, Thomas Antretter, Werner Ecker

**Affiliations:** 1Materials Center Leoben Forschung GmbH, Roseggerstraße 12, 8700 Leoben, Austria; Werner.Ecker@mcl.at; 2Voestalpine Tubulars GmbH & Co KG, Alpinestrasse 17, 8652 Kindberg-Aumuehl, Austria; Gerald.Winter@vatubulars.com (G.W.); Juergen.Klarner@vatubulars.com (J.K.); 3Institute of Mechanics, Montanuniversitaet Leoben, Franz Josef Strasse 18, 8700 Leoben, Austria; Thomas.Antretter@unileoben.ac.at

**Keywords:** simulation, XRD measurements, design, residual stress, phase transformation, low-alloyed steel

## Abstract

Residual stresses in quenched seamless steel tubes highly depend on the cooling conditions to which the tubes have been subjected. The design aspect of how to use controlled cooling strategies in multiphase steel tubes to achieve certain residual stress and phase configurations is discussed. In an experimentally validated finite element (FE) model considering a coupled evolution of martensite and bainite, three cooling strategies are tested for a low-alloyed 0.25 wt.% C steel tube. The strategies are (i) external cooling only, (ii) internal and external cooling for low residual stresses in a mainly martensitic tube, and (iii) internal and external cooling with low cooling rate for a mainly bainitic tube. The strategies represent design cases, where low residual stresses with different phase compositions are provoked, in order to show the potential of numerical analysis for residual stress and property design. It can be concluded that, for the investigated steel class, intense external cooling leads to a characteristic residual stress profile regardless of the dimension. A combination of external and internal cooling allows a more flexible design of residual stress and phase distribution by choosing different cooling parameters (i.e., water amount and cooling times). In general, lower cooling rates lead to lower thermal misfit strains, and thus less plasticity and lower residual stresses.

## 1. Introduction

The novel aspect of this work is to use an experimentally verified multiphase process model and present mechanism-based cooling strategies tailored to specific requirements, such as defined residual stress state and multiphase composition.

Residual stresses are nowadays usually calculated using finite element (FE) models. Denis et al. [1] gave a mathematical description on how to couple thermal, mechanical, and metallurgical fields including phase transformation in 1992, which is the essential framework for calculating residual stresses during heat treatment. Rhode and Jeppsson [2] published a detailed literature review on the established models and a comparison of experiment and simulation. A very general overview for numerical modelling for stresses in heat treatment applications can be found in the work of [3].

The effect of manufacturing process and material parameters on residual stresses or distortion considering phase transformation has already been investigated using FE methods for different applications. For welding, for example, Fu et al. [4] investigated in 2016 the influence of welding sequences on distortion, while Bhatti [5] focused on the influence of thermo-mechanical material properties on residual stresses and distortion. Islam et al. [6] additionally used numerical optimization to reduce distortion for arc welding processes and combined distortion modelling with optimization. For conventional hardening, Prime investigated and predicted residual stresses in a steel ring [7] and Eck and coworkers [8] modelled the residual stress evolution in a more complex component, that is, tire protection chains. Schemmel et al., in 2015, pointed out the effect of component sizes on residual stress formation [9].

When phase transformation is involved, as in ferritic steel products, the concomitant volume expansion and transformation-induced plasticity (TRIP) [10] can have a strong impact on the resulting residual stress distributions, as discussed in the work of [9]. Deng pointed out the impact of phase transformation in medium carbon steels has on residual stresses [11]. When multiple phase transformations (i.e., austenite to martensite and bainite) appear, however, this has to be accounted for in the TRIP strain and modelling strategy is proposed.

The systematic use of simulation to improve residual stresses or distortion for ferritic steels has been discussed in the literature. Especially for welding, recently published papers investigated the tuning of process parameters in order to reduce residual stresses and distortions [12]; however, in some cases without considering TRIP and metallurgical volume expansion, as in the literature on welding mentioned above from Islam, Fu, and Bhatti. Nallathambi et al. [13] investigated the systematic reduction of distortion and the effects of certain material properties such as yield strength and transformation start temperatures with the FE method for cooled steel profiles. Islam et al. [6] even used optimization techniques to minimize distortion in welds, but did not take into account phase transformation effects.

In this paper, a numerical simulation of a tube quenching process is presented considering all relevant phase transformation effects with the goal of testing sets of process parameters that result in residual stresses as low as possible. To find processing routes that result in desired residual stress states with a defined phase composition, for example, mainly martensitic, mainly bainitic, or mixed, model-based design strategies are developed and used instead of iterative optimization routines.

It can be distinguished between *minimum*, *optimum*, and *low* residual stress configurations. *Low* residual stresses (close to zero) do not always imply that this is the *optimum* stress configuration, because, in some cases, it is beneficial to have high compressive residual stresses (however, residual stresses must be always self-equilibrated, and thus compressive stresses near the surface must lead to tensile stresses elsewhere). Stress configurations showing *low* residual stresses also do not automatically imply that this is a *minimum* possible residual stress configuration. To find a global minimum for all phase transformation and cooling configurations, a numerical minimization would be necessary, as performed, for example, in the work of [6].

The focus of this work is to use a FE integrated physical-based cooling model and test different cooling strategies to design different phase distribution and stress configurations computationally. In the present work, the simulation results of three different strategies to provoke certain stress and phase configurations for a tube with 200 mm outer diameter and 22.65 mm (200 × 22.65 mm) wall thickness are investigated:

The cooling strategies include the following:(i)external cooling only, that is, the coolant is applied only to the tube’s outer surface;(ii)both, external and internal cooling for mainly martensitic microstructure, that is, an additional cooling is applied through a cooling device from inside the tube;(iii)and both external and internal cooling, mainly to adjust a bainitic microstructure.

The critical physical quantity, giving rise to residual stresses in the investigated low-alloyed steels, is the temperature distribution, which, in turn, influences plasticity and TRIP. A strategy on how to reduce residual stresses after quenching is proposed.

The main contribution of this work is modelling strain and stress evolution during coupled bainite and martensite transformation in seamless tubes. Differences between low-stress and high-stress cooling concepts are pointed out and mechanism-based cooling strategies are proposed to design tailored residual stress configurations in multiphase steel tubes.

## 2. Materials and Methods

This section gives a brief model description of the investigated material, the used FE model and the employed cooling strategies.

### 2.1. Model Description

A tube with 200 mm outer diameter and 22.65 mm wall thickness is modelled by an axisymmetric thin strip (see Figure 1) subject to generalized plane strain condition ($\varepsilon_{z}=const.$) in axial direction. The FE software ABAQUS/Standard (version 2018) [14] is used and the chosen 2D axisymmetric element types are DCAX4 for the thermal and CAX4 elements for the mechanical calculation step with a size of 0.12 × 0.12 mm and a generalized plane strain condition in axial direction. The model thickness is 0.12 mm in axial direction, that is, one element line. The mechanical boundary conditions are shown in Figure 1 and the thermal boundary conditions apply the heat transfer coefficients $\alpha_{os}$ on the outer surface and $\alpha_{is}$ on the inner surface and a continuous condition on the surfaces perpendicular to axial direction.

The tangential and radial strains $\varepsilon_{t}$ and $\varepsilon_{r}$ are
(1)$$\varepsilon_{t}=\frac{u_{r}}{r}\;\mathrm{and}\;\varepsilon_{r}=\frac{\partial u_{r}}{\partial r}$$

The cooling boundary conditions on the inner and outer surface impose the assumed process cooling conditions. An austenitic tube moves with 880 °C into an array of eight cooling baskets with short gaps between them; within one cooling basket, the heat transfer coefficient is calculated following the work of [15] as a function of temperature and water amount and is greater than in the gaps, where the tube has no water contact, but is exposed to air. For contact with air, $\alpha_{os}\;$ is constant at 40 W/m^2^K. When internal cooling is applied, $\alpha_{is}\;$ is calculated as described above using the work of [15], and is not applied by eight baskets, but by one inner cooling device. The setup is implemented in the FE software ABAQUS with the user subroutine FILM for the heat transfer coefficient, and a more detailed description of the manufacturing setup can be found in the work of [16].

The following variations of water amounts and cooling speeds are used, see Table 1 (the speed at which the tube moves along the cooling system).

The variation of the applied water amount changes the heat transfer coefficient, and thus the heat flux, which in turn governs the local temperature distributions and phase evolution.

Diffusionless, that is, martensitic phase transformation, is modelled using the Koistinen–Marburger relationship [17], where the martensite growth rate ${\dot{z}}_{m}$ is implemented proportional to the rate of undercooling $\dot{T}$ and the still transformable remaining austenite fraction $z_{\gamma }$, that is, ${\dot{z}}_{m}=-\beta \dot{T}z_{\gamma }$.

The phase transformation model for diffusive phase transformation is implemented using a model developed in three consecutive papers by Garrett and Mahnken [18] for the basic framework, and in the works of [19,20], who added a distinction for upper and lower bainite and included additional variant selection.

The model parameters were taken from the work of [16]. Alternative models for diffusive phase transformation at non-isothermal conditions use, for example, a stepwise isothermal Scheil approach based on the additivity principle described in detail in the works of [1,3]. Lusk et al. [21] consider a set of thermodynamic Arrhenius-type equations as implemented, for example, by Prime [7]. In this work, a model proposed by Mahnken was chosen for its higher accuracy over a wide cooling range and the physical foundation of most of its parameters.

The phase transformation models are implemented in an incremental form in ABAQUS using user subroutines [14]. The subroutine USDFLD assigns N – 1 calculated product phase fractions to N – 1 variables of a user defined field, as the mother phase austenite is not assigned to an individual variable. To account for the change in properties for a combination of different phases, namely, the change in thermal expansion, volume expansion, TRIP strain, latent heat, Young’s modulus, and Poisson’s ratio, the properties $P_{k}$ for *k* = 0, …, *N* phases are linearly weighted by their phase fraction $z_{k}$ by the following:(2)$$P\left(T,z_{k}\right)=\overset{N}{\underset{k=0}{\sum }}P_{k}z_{k}$$

The weighted thermal strain increment ${\dot{\varepsilon }}^{th}$ and volume expansion increment ${\dot{\varepsilon }}^{vol}$ are calculated in the user subroutine UEXPAN. The equivalent TRIP strain increment ${\dot{\varepsilon }}^{TRIP}$ is calculated in the user subroutine CREEP. The user subroutine HEATVAL accounts for the change in latent heat, following the works of [3,16].

The TRIP strain during phase transformation [10] is implemented using the formulation of the work of [22]. The formulation was slightly adapted in the work of [16] and this work to account for the fact that the austenite is consumed by multiple phase transformations. The modification was not used to the full extend in our previous work [16], as the focus was on model development and not process design, but is of relevance now for combined martensite and bainite phase transformation and reads as follows:(3)$${\dot{\varepsilon }}_{ij,k}^{TRIP}\;=\;\frac{3}{2}K_{k}S_{ij}f^{\prime }\left(z_{1}\ldots z_{N}\right){\dot{z}}_{k}$$
where $z_{k}$ is the fraction of product phase *k* and the function *f’(*$z_{1}\ldots z_{N}$*)* ensures that the corresponding TRIP strain saturates with decreasing parent phase, which has transformed to 1, …, *N* product phases:(4)$$f^{\prime }\left(z_{1}\ldots z_{N}\right)\;=\;2\;\left(1-\overset{N}{\underset{i=1}{\sum }}z_{i}\right).$$

The material parameters $K_{k}$, also known as the Greenwood–Johnson [10] parameter, are determined by means of dilatometric experiments performed at two different cooling rates to induce martensite or bainite formation, respectively, and subject to an additional mechanical load following the method as proposed in the work of [23]. These experiments also reveal if the phase transformation start temperatures ($M_{s}$ and $B_{s}$) are sensitive to the applied stress. Figure 2 shows the results for the transformation start temperatures for martensite (*λ* = 0.1) (*λ* is time required for cooling from 800 to 500 °C in hectoseconds) and bainite (*λ* = 1.1) for an applied stress ranging from −90 to 90 MPa, which is about half the yield strength of the austenite at this temperature. 

The stress applied during phase transformation has only a small impact on the transformation start temperature, as shown in Figure 2—not at all for the austenite–martensite transformation (λ = 0.1) and very little for austenite-bainite phase transformation (λ = 1.1). This allows a numerical calculation of the cooling problem in a weakly coupled manner because applied stresses do not affect the solution for the thermal field. So, the transient heat problem is solved first and the results serve as input for the mechanical problem.

The theoretical Greenwood–Johnson parameter for the martensite phase transformation at room temperature is as follows [10]:(5)$$K\;=\;\frac{5}{3}\frac{\Delta V/V}{\sigma_{Y,\gamma }}\;\sim \;\frac{5}{3}\frac{0.0261_{RT}}{250MPa}\;\approx 8.7\;\times \;10^{-5}{\;\mathrm{Pa}}^{-1\;}$$

The volume expansion ΔV/V is 0.0261 (–) after cooling to room temperature for martensite and 0.0264 (–) for bainite, respectively. The value was calculated from dilatometer curves and corresponds to volume expansions given by the work of [24] for a steel with 0.25 wt.% C. The yield strength $\sigma_{Y,\gamma }$ is the yield strength of the weaker phase, that is, the austenite yield strength of 250 MPa at a temperature range covering the transformation interval. Because it cannot be measured directly, it has to be extrapolated from tensile tests carried out at temperatures >M_S_; see the work of [25]. The measured values for the Greenwood–Johnson parameter are $K_{M}=8.9\;\times \;10^{-5}$ MPa^−1^ for martensite phase transformation and $K_{B}=8.83\;\times \;10^{-5}$ MPa^−1^ for bainite.

Temperature dependent thermo-physical properties are used, namely, the thermal expansion coefficient, heat capacity, and thermal conductivity, as well as temperature dependent elastic and plastic properties, with a temperature dependent Young’s modulus and Poisson’s ratio (calculated using JMatPro [26]) and temperature dependent yield strength and flow curves from measurement data for the phases austenite, martensite, and bainite. The material data and flow curves can be found in a previous publication [16].

For the model validation, a tube of the investigated dimension was produced in an industrial process using internal and external cooling and high-energy X-ray diffraction measurements were performed to determine tangential and radial residual stresses [16]. The process conditions were applied to the FE model and the simulation results are compared with the measurements in Figure 3. The calculated stresses correlate qualitatively and quantitatively with the measurements in tangential direction. On the inner and outer surface, radial stresses have to be zero and the observed deviation in the measurement results is the result of artefacts.

### 2.2. Design Strategy

Considering two possible phase transformations in the model (i.e., from austenite to martensite/bainite), the current work focuses on finding cooling conditions that result in residual stresses close to zero MPa and different phase constituents.

Even if compressive stresses on the outer surface as for strategy 1 can be beneficial in some applications (e.g., corrosion, crack initiation), they may be detrimental in terms of distortion in subsequent process steps, such as annealing or machining. Hence, this work focuses generally on the stress evolution and, for this particular application, on their reduction.

It has been shown by the authors of [16] that areas with severe plastic deformation in austenite owing to thermal misfits show high residual stresses at room temperature. Therefore, the approach was to lower the thermal misfit in austenite, by lowering the temperature gradients with more moderate cooling and still preserving mainly martensite phase composition, as in strategy 2. To this end, the applied water amount was reduced and adjusted individually at the inner as well as the outer surface to obtain similar temperature gradients.

One way of producing mainly a bainitic microstructure is to cool at very low rates following a almost horizontal line of a continuous cooling transformation (CCT) phase diagram. In this work, however, the goal is to get lower bainite by ‘quasi’ isothermal transformation close to the martensite start temperature, as shown in strategy 3. This is achieved by quick cooling on the surfaces combined with self-annealing and the still warm inner area of the tube and is discussed in detail in the following results section.

## 3. Results

In this section, the simulation results for three different cooling strategies are discussed. The residual stress in the radial, axial, and tangential direction are denoted as $\sigma_{r},\;\;\sigma_{z}$, and $\sigma_{t}$. The total strain ε is split up into the sum of contributions from the elastic part $\varepsilon^{el}$, thermal expansion $\varepsilon^{th}$ and metallurgical volume expansion $\varepsilon^{vol}$, and inelastic strain contributions from plasticity $\varepsilon^{p}$ and transformation induced plasticity $\varepsilon^{trip}$. The used subscripts indices then denote the respective strain component.
(6)$$\varepsilon =\;\varepsilon^{el}+\varepsilon^{th}+\varepsilon^{vol}+\;\varepsilon^{p}+\varepsilon^{trip}$$

### 3.1. Strategy 1

Cooling strategy 1 uses external cooling only, which for the investigated dimension, 200 × 22.65 mm, leads to a mainly martensitic microstructure. Figure 4a shows the temperature evolution at a point on the inner surface and at the same axial position on the outer surface of the tube. The spikes of the curve pertaining to the temperature evolution on the outer surface result from the setup of the cooling equipment. The minima in the temperature distribution are always at the end of a cooling basket. Between two baskets, the still hot inner regions re-heat the outer surface when there is no contact with water, leading to the characteristic spiked temperature evolution.

Figure 4b shows the residual stress distribution and plastic strain contributions $\varepsilon_{t}^{p}$ and $\varepsilon_{t}^{trip}$ in the tangential direction as a function of the distance from the outer surface. The compressive stresses in the tangential *σ_t_* and axial *σ_z_* direction of ~900 MPa decaying towards the inner surface are characteristic for external cooling. Comparing this stress distribution for the dimension of 200 × 22.65 mm with a smaller dimension of 177.8 × 12.65 mm investigated in the work of [16] reveals that this trend is independent of the tube’s dimension for the given steel class and cooling intensity.

The residual stress distribution in tangential and axial direction is essentially determined by the inelastic strain contributions $\varepsilon^{p}$ and $\varepsilon^{trip}$. Thermal misfits that arise during cooling lead to plastic strain in austenite, which influences the stress distribution during transformation, and thus affects the TRIP contributions, $\varepsilon^{trip}$. Meanwhile, the volume jump due to transformation into either martensite or bainite always leads to similar strain contributions $\varepsilon^{vol}$ over the cross section, and is thus not depicted in Figure 4b. The evolution of plastic strain and TRIP strain for a purely martensitic, but representative reference case is discussed in detail in the work of [16].

Figure 5a shows snapshots at specific times of the distribution of the martensite (green) and bainite (blue) fraction as a function of the distance from the outer surface. Figure 5b shows the front view of Figure 5a. The plots show how the phase transformation progresses; the martensitic transformation starts at the outer surface and progresses towards the inner surface over time. A maximum of 20% bainite forms in the middle region.

The bainite formation in this middle region happens as a result of the cooling setup; here, the material points transform partly to martensite, but the transformation stops between two cooling baskets owing to reheating, which raises the temperature above the martensite start temperature $M_{s}$. Owing to the preceding transformation history (microplasticity, interface formation, and so on), the remaining austenite contains a large number of nucleation sites, thus giving rise to an accelerated formation of bainite. This is accounted for by the term (1 − *z*)*^γ^* in Equation (12) the work of [19]. The variable γ is a constant growth parameter and z is the already transformed product phase fraction. 

The expression (1 − *z*)*^γ^* was modified in this work to account for the multiphase transformation to ${\left(1-\overset{N}{\underset{i=1}{\sum }}z_{i}\right)}^{\gamma \;}\;$ for I = 1, …, *N* product phases. This reduces the remaining austenite fraction also by the previously formed martensite fraction, which is no longer available for bainite formation.

### 3.2. Strategy 2

Figure 6a,b show the temperature distribution for a quenching concept generating low-residual stresses; in this strategy, internal cooling is applied additionally to the external cooling (temperature on the inner surface is the green, dashed line in Figure 6a). Martensite transformation starts from both surfaces: inner and outer. The inner region is cooled for only 3 s by an inner cooling device inserted into the tube, and is re-reheated subsequently by the still hot middle region of the tube’s wall (curves for several different points in the middle region are not shown for better readability).

Comparing Figure 6b to Figure 4b shows that the residual stresses are reduced significantly; for example, on the outer surface from −900 Mpa to −200 Pa and on the inner surface form 20 Mpa to −150 Mpa.

A comparison of the distribution of the plastic strain component in tangential directions shown in Figure 6b versus Figure 4b reveals the effect of the temperature gradient. These thermal misfits cause classical plasticity in the austenite with low yield strength.

The TRIP strain $\varepsilon_{t}^{trip}$ remains of a similar magnitude (in the range of −0.4% to 0.0%) as for strategies 1 and 2, despite the slightly slowed down transformation—but with different distribution owing to the altered temperature profile and thermal misfits.

This is an expected result, as the austenite yield strength is not implemented as strain rate dependent, because no strong strain rate dependence was detected experimentally—and the integral under the phase evolution rate integrated over time must always be 1. Thus, a fully completed phase transformation always leads to a similar TRIP magnitude for similar cooling rate and the distribution is altered owing to changes in the stress state during transformation.

Only the stress configuration changes as a result of thermal misfits. When the thermal misfits differ between the two cooling strategies, they change the stress state, plastic strain, and TRIP strain.

Comparing Figure 7b with Figure 10 in the work of [16] shows that a mainly martensitic microstructure is achieved in both cases. In the present study, much lower residual stresses are obtained by reducing the cooling intensity to a minimum.

### 3.3. Strategy 3

The effect of the misfit strains in austenite becomes more obvious in cooling strategy 3, where martensite is formed only near the inner and the outer surface, while 100 vol.% bainite is formed in the middle region. The bainite formation is more favored in strategy 3 by the slow cooling in the middle region, as compared with strategy 1 and strategy 2; see Figure 6a and Figure 8a. Between 100 s and 300 s, bainite forms ‘quasi-isothermal’ with a cooling rate of about 0.5 K/s. When the cooling rate becomes lower after 70 s, and the ‘quasi-isothermal’ transformation begins. The radial temperature gradient across the tube’s wall is close to zero; see Figure 8a.

The plastic strain in tangential direction of 0.5% caused by thermal misfit on the outer surface (see Figure 8b) is comparable to the plastic strain in strategy 2, as the cooling rates near the surface are initially similar. However, because, in strategy 3, the total amount of cooling is lower, the martensite transformation stops after 6 mm from the outer and 1.5 mm from the inner surface.

Between 5 mm and 10 mm, the plastic strain drops to a lower value of 0.2%, from 0.5% to 0.2%. Comparing this to Figure 9a reveals that this corresponds to the phase change caused by the change in the cooling conditions.

This overall bainite evolution for this ‘quasi-isothermal’ condition can be seen in Figure 9b, where the phase fraction evolution follows an S-shape in a time interval between 50 and 350 s, as expected for a diffusive phase transformation.

As mentioned in the previous section, the volume expansion of bainite formation is slightly larger than for martensite transformation (0.261% for martensite compared with 0.264% for bainite). The results show that the change in the residual stresses results from the much lower cooling rate, leading to less plastic strain and lower stresses during transformation. This in turn changes the TRIP strain and leads to overall residual stresses lower than 200 MPa.

The presented results refer to a low-alloyed steel class with 0.25 wt.% carbon. Changes in the composition and grain size can affect the transformation start temperatures, as shown by the authors of [27]. Especially carbon affects the resulting yield strength, as documented experimentally by the authors of [28]. Volume expansion, yield strength, transformation start temperatures, and kinetics have been systematically shown to affect the resulting distortions or stresses by the authors of [5]. The plastic strain and TRIP strain are dependent on the austenite yield strength, which, in turn, correlates to the austenite grain size according to the Hall–Petch relationship, as shown by the authors of [25]. This implies that, when the austenite grain size is reduced, the yield strength and thus the resulting TRIP strain and plastic strain distribution will be lowered. The martensite yield strength depends on the block size and also corresponds to the former austenite grain size, as shown by the authors of [29].

## 4. Discussion and Conclusions

A multiphase numerical model is presented to show three systematic design strategies on how to adjust distinct microstructures and stress states in low-alloyed seamless steel tubes. The model accounts for the different physical phase transformation mechanisms relevant at the respective cooling rates. Three distinct cooling strategies and their effect on phase fraction and residual stress evolution were investigated: (i) strategy 1, resulting in a mainly martensitic microstructure; (ii) strategy 2, generating very low residual stress; and (iii) strategy 3, resulting in a predominantly bainitic microstructure.

The use of intense external cooling leads to a characteristic residual stress distribution regardless of the geometry with compressive stresses on the outer surface (177.8 mm × 12.65 mm [16] or 200 mm × 22.65 mm) for the investigated material and wall thickness range. Depending on the application and subsequent processing steps, a low-stress concept is not always a desirable option. If compressive stresses are favored, the simulation framework can be employed to find a maximum for compressive stresses on the inner and/or outer surface.

A combination of internal and external cooling can lower residual stresses considerably, while a similar microstructure can be preserved. This is shown by the comparison of strategy 1, using external cooling only, and strategy 2, where a combination of internal and external cooling and lower cooling rates reduce thermal misfits. The thermal misfits cause plastic strain $\varepsilon^{p}$ in the austenitic state, as can be best seen in strategy 1, which is one of the main contributors to residual stresses. For the investigated cooling cases, the resulting residual stresses are dominated by both plastic strain, $\varepsilon^{p},$ and TRIP strain,$\;\varepsilon^{TRIP}$.

Different microstructures, mainly martensitic or mainly bainitic, at low residual stress levels can be adjusted with different intensities of internal and external cooling. Rapid cooling on both surfaces, as in strategy 2, can yield mainly martensitic microstructure, while short cooling followed by ‘quasi-isothermal’ bainitic transformation, as in strategy 3, yields a mainly bainitic microstructure, but both strategies result in low residual stresses.

To reduce residual stresses effectively, balanced phase transformation fonts from both surfaces during quenching are favorable and temperature gradients should be as low as possible to keep thermal misfits low, while still adjusting the desired microstructure. Further research should be dedicated to investigating the fundamental effect of prior martensite formation on the subsequent bainite transformation kinetics experimentally, with simulation methods, and to extending the modelling framework to include changes in chemistry and austenite grain sizes.

## Figures and Tables

**Figure 1 materials-13-00439-f001:**
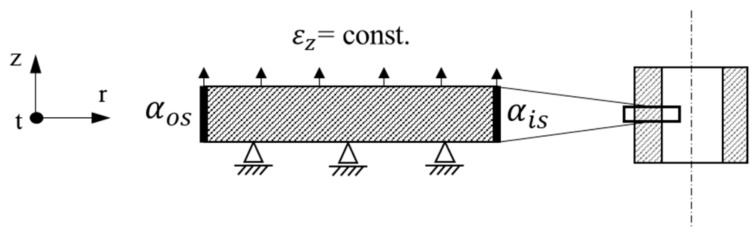
Schematic depiction of model and the applied boundary conditions.

**Figure 2 materials-13-00439-f002:**
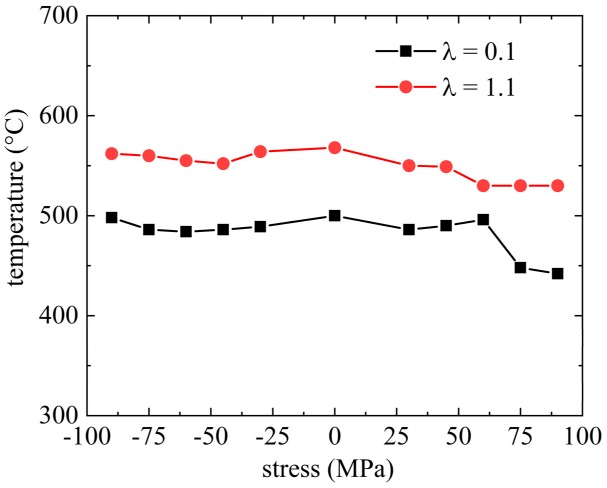
Dependency of the transformation start temperature of martensite $M_{s}$ (blue, λ = 0.1) and of bainite $B_{s}$ (red, λ = 1.1) for different applied stresses.

**Figure 3 materials-13-00439-f003:**
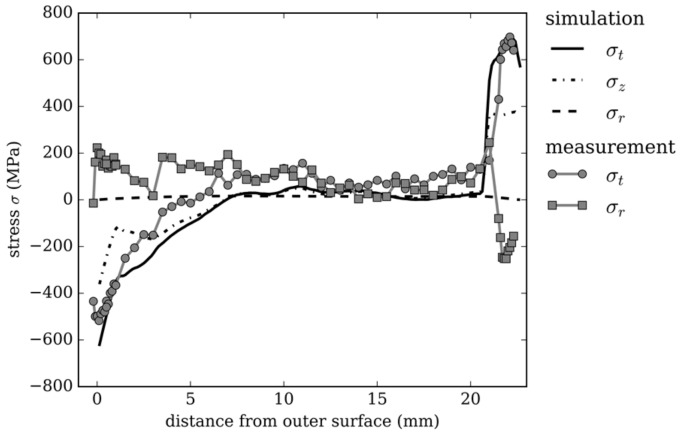
Comparison of high-energy X-ray diffraction residual stress measurements with simulation results for an industrial case.

**Figure 4 materials-13-00439-f004:**
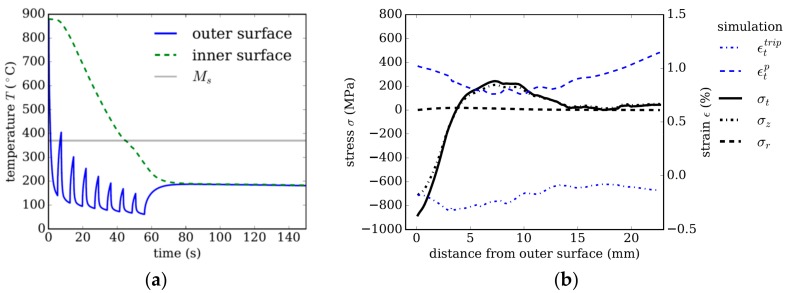
Strategy 1. (**a**) Temperature evolution using external cooling only. (**b**) Resulting residual stresses and plastic strain in tangential direction after quenching as a function of distance from the tube’s outer surface.

**Figure 5 materials-13-00439-f005:**
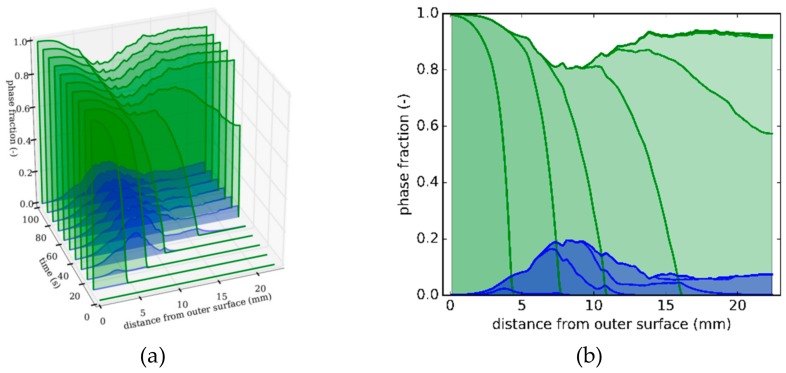
Strategy 1. (**a**) Snapshots at different times of phase distributions over distance from outer surface (green: martensite; blue: bainite). (**b**) Front view of Figure 5a depicting phase distributions as a function of distance from the tube’s outer surface.

**Figure 6 materials-13-00439-f006:**
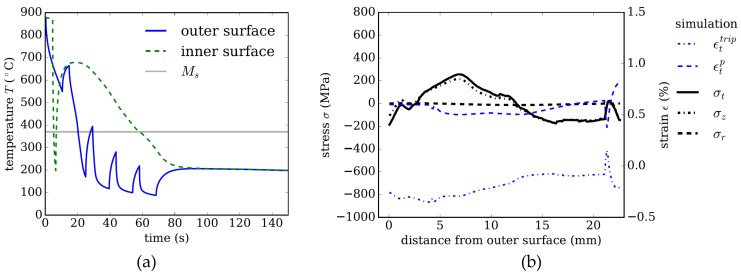
Strategy 2. (**a**) Temperature evolution using external and internal cooling. (**b**) Resulting residual stresses and plastic strain in tangential direction after quenching as a function of distance from the tube’s outer surface.

**Figure 7 materials-13-00439-f007:**
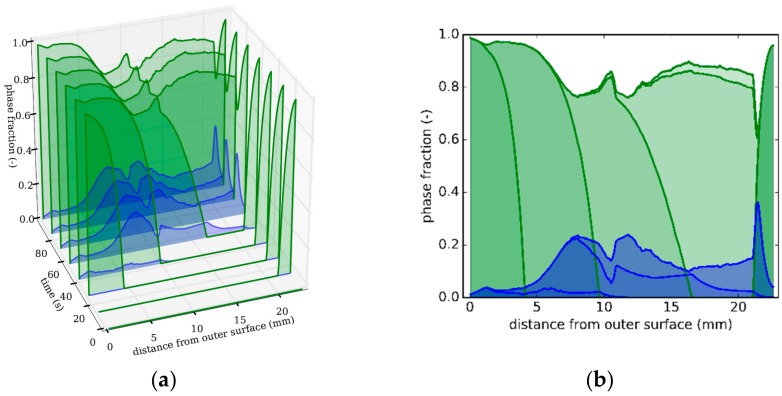
Strategy 2. (**a**) Snapshots at different times of phase distributions over distance from outer surface (green: martensite; blue: bainite). (**b**) Front view of Figure 7a depicting phase distributions as a function of distance from the tube’s outer surface.

**Figure 8 materials-13-00439-f008:**
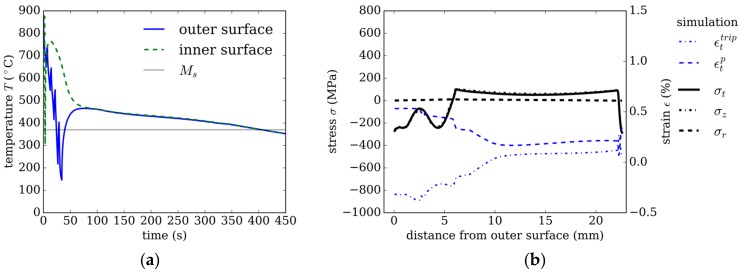
Strategy 3. (**a**) Temperature evolution using internal and external cooling. (**b**) Resulting residual stresses and plastic strain in tangential direction after quenching over distance from the outer tube’s surface.

**Figure 9 materials-13-00439-f009:**
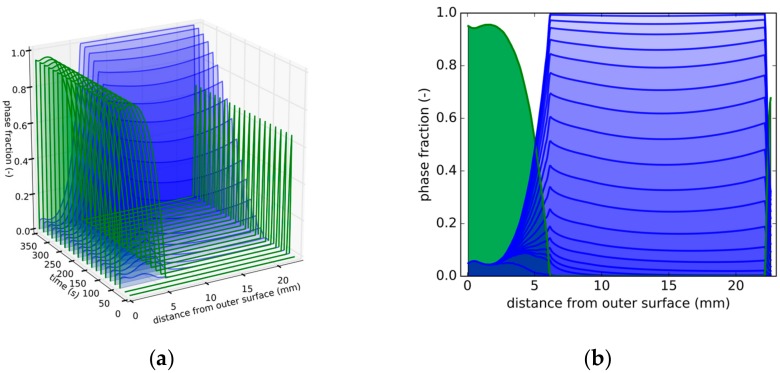
Strategy 3. (**a**) Snapshots at different times of phase distributions over distance from outer surface (green: martensite; blue: bainite). (**b**) Front view of Figure 9a depicting phase distributions as a function of distance from the tube’s outer surface.

**Table 1 materials-13-00439-t001:** Cooling parameters for the three different cooling strategies.

Title	Internal Cooling	External Cooling	Cooling Speed
	(kgm^−2^s^−1^)	(kgm^−2^s^−1^)	(ms^−1^)
Strategy 1	-	80	0.2
Strategy 2	100	10	0.2
Strategy 3	100	10	0.1

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
