# Peer review of "Model-Based Residual Stress Design in Multiphase Seamless Steel Tubes"

_materials, 2020, doi:10.3390/ma13020439_

Round 1

Reviewer 1 Report

In fig. 2, in abscissa, must to be, STRESS, not stess!

You must to have CONCLUSIONS

Reviewer 2 Report

Manuscript number Materials-656777
Title: Model based residual stress design in multiphase 2 seamless steel tubes

The manuscript presented the a multiphase numerical model of seamless steel tubes under three distinct cooling strategies to achieve different residual stresses and phase configurations.

This is an extended work from previous numerical simulation studies on residual stress and microstructure evolution in steel tubes for different cooling conditions [15].
Overall the proposed methodology are sound and results are interesting. However, there are many parts that are not clearly written in English that affect the quality of the paper.

The whole manuscript needs to be revised as there are many mistakes in English and many parts have not got clear meaning and required clarification. In addition, a summary of the finite element simulation setup with key information on meshing, process modelling, thermal and mechanical boundary conditions, data input and measurements should be included.

For example, some detailed comments on mistakes in English are as follows:

1- The paper should be written in third person, i.e. line 14 in the Abstract "...We discuss...", line 70 "...We distinguish...", line 79 "...we investigate..." etc.
2- Lines 14-15: "We discuss the design aspect how to use controlled cooling strategies in multiphase steel tubes to achieve certain residual stress ..." should be "We discuss the design aspect OF how to use... to achieve certain residual stressES ..."
3- Lines 18-20: It is not clear about this "The strategies use (i) external cooling only, (ii) internal and external cooling for low residual stresses in a mainly martensitic tube and (iii) internal and external cooling with low cooling rate for a mainly bainitic tube."
4- Line 22: " It can be concluded, that..." should be " It can be concluded that..."
5- Lines 24-25: "...a broad field of variation to control residual stresses..." a broad field of variation IN WHAT?
6- Lines 34-35: "Residual stresses are nowadays frequently calculated using finite element (FE) models. Denis et al. gave a mathematical description..." use "usually" or "often" instead of "frequently". Add reference [1] to the end of Denis et al.
7- Line 44: Add reference [6] to the end of Islam et al.
8- Line 52: Remove e.g.
9- Line 57: "Especially for welding recently published papers..." use "," after welding - shoud be " Especially for welding, recently published papers ..."
10- Lines 63-64: "...but without taking into account phase transformation effects" should be "but DID NOT TAKE into account phase transformation effects"
11- Lines 67-69: It is not clear about this " Rather than using classical optimization techniques, we employ engineering design considerations with the objective to identify a processing route resulting in a desired residual stress state given a defined phase composition."
12- Line 73: "... residual stresses do also not automatically..." should be "... residual stresses ALSO DO NOT automatically..."
13- Line 75: What does this mean "a defined residuum"
14- Line 88: "A strategy how to reduce..." should be "A strategy ON how to reduce..."
15- Line 153- Figure 2: "Stess (MPa)" should be "Stress (MPa)"

[15] Brunbauer, S.; Winter, G.; Antretter, T.; Staron, P.; Ecker, W. Residual stress and microstructure evolution in steel tubes for different cooling conditions – Simulation and verification. Materials Science and Engineering: A 2019, 747, 73–79.

Reviewer 3 Report

This paper studies the residual stresses in quenched seamless steel tubes using FEA. Three strategies including external cooling only, (ii) internal and external cooling are considered. This paper is well orginized and well written. Two revision requirments:1. how to implement the phase transformation model using FEA should be described in detail. 2. Temperature-dependent material parameters should be considered.  

Round 2

Reviewer 3 Report

I recommend the publication of this paper on Materials.